# *Nicotiana noctiflora* Hook. Genome Contains Two Cellular T-DNAs with Functional Genes

**DOI:** 10.3390/plants12223787

**Published:** 2023-11-07

**Authors:** Galina V. Khafizova, Nicolas Sierro, Nikolai V. Ivanov, Sofie V. Sokornova, Dmitrii E. Polev, Tatiana V. Matveeva

**Affiliations:** 1Department of Genetic and Breeding, St. Petersburg State University, Saint Petersburg 199034, Russia; galina.khafizova@gmail.com (G.V.K.);; 2PMI R&D, Philip Morris Products S.A., Quai Jeanrenaud 5, CH-2000 Neuchâtel, Switzerland; nicolas.sierro@cgat.ch (N.S.); nikolai.ivanov@unine.ch (N.V.I.); 3St. Petersburg Pasteur Institute, Saint Petersburg 197101, Russia

**Keywords:** *Nicotiana noctiflora* Hook., cT-DNA, horizontal gene transfer

## Abstract

*Agrobacterium (Rhizobium)*-mediated transformation leads to the formation of crown galls or hairy roots on infected plants. These effects develop due to the activity of T-DNA genes, gathered on a big plasmid, acquired from agrobacteria during horizontal gene transfer. However, a lot of plant species are known to contain such sequences, called cellular T-DNAs (cT-DNAs), and maintain normal phenotypes. Some of the genes remain intact, which leads to the conclusion of their functional role in plants. In this study, we present a comprehensive analysis of the cT-DNAs in the *Nicotiana noctiflora* Hook. genome, including gene expression and opine identification. Deep sequencing of the *Nicotiana noctiflora* genome revealed the presence of two different *c*T-DNAs, *Nn*T-DNA1 and *Nn*T-DNA2, which contain the intact genes *iaaM*, *iaaH*, *acs*, *orf13*, *orf13a*, and *orf14*. According to the expression analysis results, all these genes are most active in roots in comparison with other organs, which is consistent with data on cT-DNA gene expression in other plant species. We also used genetic engineering approaches and HPTLC and HPLC-MS methods to investigate the product of the *acs* gene (agrocinopine synthase), which turned out to be similar to agrocinopine A. Overall, this study expands our knowledge of cT-DNAs in plants and brings us closer to understanding their possible functions. Further research of cT-DNAs in different species and their functional implications could contribute to advancements in plant genetics and potentially unveil novel traits with practical applications in agriculture and other fields.

## 1. Introduction

*Agrobacterium (Rhizobium)*-mediated transformation is currently the main method for obtaining transgenic plants and one of the key methods for delivering components for genome editing into plant cells. These techniques are based on the ability of agrobacteria to integrate a fragment of their Ti or Ri plasmid, called T-DNA, into the plant genome [1,2,3]. T-DNA genes have eukaryotic regulatory elements, so they are expressed in plant cells. In the case of transformation by wild-type plasmids, the expression of T-DNA genes leads to the abnormal growth of transgenic plant tissues on non-transgenic plants and to the synthesis of small molecules called opines [4]. Opines can be metabolized as a source of carbon and nitrogen for the nutrition of the same strains of *Agrobacterium* sensu lato [4,5]. At the same time, there are plants found in natural conditions, containing T-DNA fragments in their genomes and transferring these sequences to their progeny. Such T-DNA sequences are called cellular T-DNAs (cT-DNAs) [6,7,8,9]. Most likely, they are descendants of plants regenerated from transgenic tissues since it has been experimentally demonstrated that, in some cases, fertile plants can be regenerated from hairy roots [10,11,12].

Initially, cT-DNA sequences were discovered in *Nicotiana glauca* (*N. glauca*). Later, more natural GMOs (nGMOs) were discovered in other representatives of the genus *Nicotiana* [6,7,8], *Linaria* [13], *Ipomoea* [14,15], and four dozen other species, whose ancestors underwent *Agrobacterium (Rhizobium)*-mediated transformation during their evolution [16,17,18,19].

Detailed evolutionary studies on cT-DNAs based on the deep sequencing of the genomes of several *Nicotiana* species of the section *Tomentosae*, subgenus *Tabacum* [20,21], revealed five different cT-DNA sequences, called TA to TE. TA-TE inserts result from independent transformation events during the evolution of the section *Tomentosae* and occur in different combinations among different species of this section: TA-TD are found in *Nicotiana tomentosiformis* (*N. tomentosiformis*), an ancestor of *N. tabacum*, i.e., tobacco; TC and TE are identified in *N. otophora*; TC, TB, and TD are described in *N. tomentosa*; and TC, TB, TD, and TA are in *N. kawakamii*. In the section *Nicotiana*, the only species in which cT-DNA sequences were studied is *N. tabacum.* It contains TA, TB, and TD inserts. Probably, TC was lost during its evolution [20]. These cT-DNAs are different from the *N. glauca* cT-DNA (section *Noctiflorae*, subgenus *Petunioides* [22]), which is referred to as gT [14]. Deep sequencing of the *N. glauca* genome confirmed the structure of the previously described gT insertion and demonstrated the absence of other types of cT-DNAs [23]. However, additional cT-DNA sequences in plants from the section *Noctiflorae* were found in transcriptome data of *Nicotiana noctiflora* Hook. (*N. noctiflora*), showing the widespread occurrence of natural transformation in this genus [24,25]. However, the structure of T-DNA and its functions in *N. noctiflora* plants have remained unexplored to date.

There is no doubt that many cT-DNA genes remain intact and are expressed [9,14,16,26,27,28,29,30,31,32]. At the same time, the set of intact genes expressed is species-specific. This means that cT-DNAs do not have a single function [33]. The function of some genes has been demonstrated at the RNA level and at the level of the relevant metabolite. Thus, some accessions of the *N. tabacum* [31], *Cuscuta* [16], and tobacco transformed by *Ipomoea acs* gene [33] have been shown to produce opines, but the biological significance of this is not yet clear.

To understand the role of *Agrobacterium*-mediated transformation in plant evolution, a detailed description of the transgenes in nGMOs, a description of the combinations of intact genes in each species, the features of accumulation of mutations, and the expression pattern are necessary. Based on genomic sequencing data, the present work reconstructs the structure of *N. noctiflora* cT-DNA, demonstrates the tissue-specific expression of its intact genes, and describes the structure of the agrocinopine encoded by the Nn *acs1* gene.

## 2. Results

### 2.1. Genome Assembly

The *N. noctiflora* genome assembly consists of 14,266 scaffolds, with an N50 length of 222,777 bp and a total length of 2,093,308,743 bp. The assembly quality value (QV) was evaluated to 33.5, and the coverage to 1.066, using Yak with a 31-mer profile generated from Illumina paired ends. The BUSCO evaluation of the assembly completeness showed the presence of 2,191 complete BUSCOs (94.2%), with 1,654 of them in a single copy (71.1%). Additionally, 28 BUSCOs (1.2%) were fragmented, and 107 BUSCOs (4.6%) were missing from the assembly.

### 2.2. cT-DNA Maps

Using the BLAST algorithm (NCBI, http://www.ncbi.nlm.nih.gov, accessed on 1 May 2023), two scaffolds containing two distinguished cT-DNAs were found in the *N. noctiflora* genome. Both the cT-DNAs are organized as imperfect inverted repeats, which is typical for cT-DNAs in *Nicotiana* genomes [8,20]. The first cT-DNA, denoted as *Nn*T-DNA1 (Acc. number OR134233.1), is 21,787 bp long. Each arm of *Nn*T-DNA1 contains ORFs showing significant homology to the agrobacterial genes *iaaH*, *iaaM*, *vis*, *mis*, *orf14*, *orf13a*, *orf13*, and *rolC*. The *acs* homolog is located in the central part of *Nn*T-DNA1 (see Figure 1a; Table 1). The second cT-DNA, *Nn*T-DNA2 (Acc. number OR134234.1), is 14,611 bp long and contains homologs of the *acs*, gene *C*, and *iaaM* genes in each of the arms (see Figure 1b; Table 1). The right arm carries a truncated version of *iaaM* (161 instead of 777 AA), and the left arm carries a truncated version of *acs* (330 instead of 372 AA). The *Nn*T-DNA1 repeat divergence is 1%, while for *Nn*T-DNA2 it is 2%.

### 2.3. Localization Sites

*Nn*T-DNA1 in the genome of *N. noctiflora* and gT in the genome of *N. glauca* carry structurally identical fragments that include homologs of the genes *rolC*, *orf13*, *orf13a*, *orf14*, and *mis*. The similarity between these fragments in *Nn*T-DNA1 and gT at the nucleotide level is 83%. Since *N. noctiflora* and *N. glauca* are evolutionarily close species, it can be assumed that *Nn*T-DNA1 and gT could be the result of a single transformation event that occurred in the common ancestral form prior to species divergence. However, the comparative analysis of contigs containing plant–bacterial boundaries in *N. glauca* and *N. noctiflora* revealed that the cT-DNAs of these species are localized at different sites in their genomes. The differing localization sites, as well as the differences in the composition of *Nn*T-DNA1 and *Nn*T-DNA2 from gT, indicate that *N. noctiflora* and *N. glauca* underwent independent transformation events during their evolutionary history, rather than acquiring cT-DNAs from a common ancestral form prior to their divergence.

### 2.4. Comparative Analysis of Nn T-DNA and Agrobacterium (Rhizobium) Homologs

Comparison of the cT-DNAs in *N. noctiflora* with all known *Agrobacterium (Rhizobium)* homologs revealed the following pattern. Different regions of *Nn*T-DNA1 demonstrate similarities with different bacterial strains. The *rolC-mis* region in *Nn*T-DNA1 is similar to the T-DNA from *Agrobacterium rhizogenes* (*A. rhizogenes*) strain 1724 (76–86%), whereas the *iaaH* and *iaaM* sequences show 74% identity to the *Agrobacterium tumefaciens* (*A. tumefaciens*) strain CFBP1935. The translated sequence of the *acs* gene in the middle of *Nn*T-DNA1 shows 82% protein identity with a hypothetical protein from *A. rhizogenes* (GAJ95539.1). A similar pattern emerges with *Nn*T-DNA2, as different segments exhibit resemblances to distinct strains of *Agrobacterium* (*Rhizobium*). Thus, while *acs-C* fragments show identity to the T-DNA from *A. rhizogenes* strain 1724, the *iaaM* gene in both arms demonstrates 68% similarity to the corresponding sequence in *A. tumefaciens* strain CFBP1935. Previously, a similar picture was shown for TB in *N. tomentosiformis* [20] and TE in *N. otophora* [21].

We identified homologs of proteins encoded by agrobacterial T-DNA genes for each gene present in *Nn*T-DNA1 and *Nn*T-DNA2, and the data are summarized in Table 1.

### 2.5. Expression of cT-DNA Genes

Analysis of the T-DNA structure has shown that most of the ORFs in *N. noctiflora* cT-DNA are interrupted by stop codons, but a few are intact and thus may retain biological activity. These are *iaaM*, *acs*, *iaaH*, *orf13*, *orf13a*, and *orf14* in *Nn*T-DNA1, and *iaaM* in *Nn*T-DNA2. Functional analysis for these genes in *Nn*T-DNA1 and *Nn*T-DNA2 was carried out using reverse-transcription real-time PCR. Expression analysis was performed for the leaves, nods, internodes, and roots of aseptically grown plants using *gapdh* as a reference gene because its expression level was stable in our experimental conditions. The data are not shown for the *orf13a* gene since its expression was lower than the reference gene in all the studied tissues. For the other genes, the relative levels of expression were calculated using the 2^−ΔΔCT^ method [34]. The highest levels for all genes were detected in the roots (Figure 2).

### 2.6. Opine Structure

The high-performance thin-layer chromatography (HPTLC) and high-performance liquid chromatography–mass spectrometry (HPLC-MS) profiles of chemical compounds differed in the control and transgenic *Escherichia coli* (*E. coli*) strains containing the opine synthase gene under the isopropylthio-β-galactoside (IPTG) inducible promoter. In the culture liquid and methanol extracts of transgenic *E. coli* cells, a significant level of a compound with sugar residues (retention time: 8.78) was detected, and it was absent in the controls. The fragmentation patterns in positive-ion MS were used to characterize the molecular structures of this compound. The most abundant characteristic fragments observed in agrocinopine-like compounds from the cultural liquid were 577.1500, 559.2999, 541.0000, and 253.1000 (Appendix A).

The major fractions, characteristic of the transgenic strain after IPTG induction, were taken for further nuclear magnetic resonance (NMR) analysis. Analysis via two-dimensional ^1^H and ^32^P NMR spectroscopy confirmed the phosphodiester bond presence (δ = 0.04) (Appendix A). All biochemical data showed good agreement with the data reported for agrocinopine A [35,36,37,38,39,40]. Thus, the substance produced by NnAcs in *E.coli* is agrocinopine A.

## 3. Discussion

In recent years, many natural GMOs have been described [16]. Based on the data available on a number of nGMOs among dicots with sequenced genomes [16,18], it can be estimated that about 7% of their genomes contain traces of *Agrobacterium*-mediated transformation. Cellular T-DNAs differ in length and number of intact genes. Extended regions of cT-DNAs are of great interest as they allow phylogenetic studies and investigations of the role of the expressed genes in plant evolution [32]. At least six cT-DNAs were described previously among representatives of the genus *Nicotiana* [20,21]. In this study, we have characterized two new cT-DNAs in the genome of *N. noctiflora*. The related species *N. glauca* also contains cT-DNA. However, its structures and localization site differ from *Nn*T-DNA1 and *Nn*T-DNA2, proving independent acts of transformation of their ancestors.

The different degree of sequence divergence within the *N. noctiflora* cT-DNA repeats indicates the different age of these insertions. A similar scenario, involving multiple acts of transformation over a period of more than a million years, has been described earlier for representatives of another evolutionary branch of *Nicotiana*, including *N. tomentosiformis* and related species [20,21]. These results together indicate the propensity of various representatives of the studied genus to *Agrobacterium*-mediated transformation.

Analysis of the fine structure of cT-DNA makes it possible to trace its phylogenetic relationships with the T-DNAs of various nGMOs and strains of *Agrobacterium/Rhizobium*.

Notably, both T-DNAs of *N. noctiflora* contained gene sets unusual for the known strains of *Agrobacterium/Rhizobium*. *Nn*T-DNA1 contains opine synthase genes that have not been previously found together. *Nn*T-DNA2 is shorter, with its set of genes truncated, and has not been described in this combination in *Agrobacterium* sensu lato. However, the detection of intact T-DNA gene sequences may indicate their functional role. We have shown that the *iaaM*, *iaaH*, *acs*, *orf13*, *orf13a*, and *orf14* genes are expressed in young plants of the studied species. The highest expression levels of all the genes were detected in the root tissues, which is consistent with the pattern previously described for other nGMO species [14,21].

In the literature, it has been described that these genes retained their functionality in other nGMOs as well. For instance, gene *C* was found in cT-DNAs such as *Ib*T-DNA1 in *Ipomea batatas* (*I. batatas*), where it remains intact [14], and TC in *N. tomentosiformis* [20]. There is a lack of information about its functioning in plants, though some impact on morphological reactions was documented. Thus, gene *C* from the *A. tumefaciens* strain C58 is known to induce shoot growth [41]. The *iaaH* and *iaaM* genes together encode indole acetic acid synthesis, which is one of the main plant hormones responsible for plant growth and development. TE in *N. otophora* [21] and *Ib*T-DNA1 in *I. batatas* contain these genes. The *iaaH* and *iaaM* in *Ib*T-DNA1 are intact and are most actively expressed in roots [14]. It is possible that the activity of *iaaM* and *iaaH* was vital right after a transformation event to stimulate root growth and a hairy root formation, giving the transformed plant an advantage and increasing its chances of survival. The *acs* gene encodes enzymes involved in the synthesis of agrocinopines. Opine synthesis genes make up the majority of cT-DNAs described to date [16,18], and it is the opine synthesis genes that often remain intact. Their expression was detected in the roots as well as in the shoots [8,14,31,42]. Moreover, detectable levels of DFG, mikimopine, and agrocinopine have been found in *N. tabacum* [31], *Cuscuta suaveolens* [17], and transgenic tobacco with the *I. batatas acs* gene [33], respectively. In our study, we also demonstrated that agrocinopine synthase from *N. noctiflora* can catalyze the production of agrocinopine A.

It has been previously shown that the opines secreted by transgenic plants can affect their microbial communities [43]. We suggest that the increase in root mass and the ability to change their biological environment due to the expression of opines could be the key factors for the fixation of cT-DNAs in the genome of *N. noctiflora*.

## 4. Materials and Methods

### 4.1. Plant Material

*N. noctiflora* cv. TW89 aseptic plants were used for DNA and RNA isolation. Seeds were obtained from the US Nicotiana Germplasm Collection, NC, USA. Seeds were germinated in vitro on Murashige and Scoog (MS) medium [44] with 80 g/L sucrose at 23 °C in a dark room until the emergence of seedlings, which were transferred to MS medium with 20 g/L sucrose, where plants were cultivated aseptically at 23 °C and under a photoperiod of 16 h day/8 h night.

### 4.2. DNA and RNA Isolation

#### 4.2.1. DNA Isolation and Preparation of Libraries

To prepare Illumina libraries, high-molecular-weight DNA was extracted from fresh leaves of *N. noctiflora* plants cultivated aseptically in vitro, according to a modified CTAB protocol for DNA macro isolation from plant tissue [45]. The protocol change is that 10% CTAB was used instead of the usual 2% CTAB. For DNA extraction young leaves were selected (mainly the second and third leaves counting from the top of the shoot). A TruSeq Nano DNA LT kit (Illumina, San Diego, CA, USA) was used for the library preparation according to the manual.

For Oxford Nanopore sequencing, high molecular weight DNA was extracted from fresh leaves of aseptic *N. noctiflora* plants using a Promega Wizard HMW DNA Extraction Kit (Promega, Madison, WI, USA) according to the manual.

#### 4.2.2. RNA Isolation

*N. noctiflora* aseptic plants were used for RNA extraction. Three plants in the vegetative stage were selected for material isolation. All plants were approximately the same size and were in equally healthy conditions in terms of the color of their leaves and the absence of signs of wilting. For each plant, a total of 4 RNA samples were isolated: from the leaves, roots, nodes, and internodes. To ensure uniformity in data collection, all samples were gathered under the same conditions with a time difference of no more than 10 min between each individual plant. Immediately after collection, the samples were flash-frozen in liquid nitrogen. Each sample was triturated in liquid nitrogen with stainless steel beads using a TissueLyser (Qiagen, Hilden, Germany) for 30–60 s. RNA from leaves, roots, nodes, and internodes was extracted using an RNeasy Plant Mini Kit (Qiagen, Hilden, Germany) according to the manual. The purity and quantity of the resultant RNA in the samples were measured using a spectrophotometer, NanoDrop 2000/2000c (Thermo Fisher, Waltham, MA, USA).

### 4.3. NGS

#### 4.3.1. Genome Assembly

DNA from aseptic *N. noctiflora* plants was used to generate Oxford Nanopore (Oxford, UK) long-reads for de novo assembly. Additionally, a paired-end library was sequenced on an Illumina HiSeq4000 (San Diego, CA, USA) for ont reads correction to increase the accuracy of the assembly.

Oxford Nanopore reads were base called using guppy 6.3.7 with the high-accuracy model, and reads with a quality lower than 7 or which were shorter than 2500 bp were discarded using seqkit [46] version 2.2.0. The genome was assembled using flye 2.9.1 [47] with the nano-hq preset and an error rate of 0.05 and polished using fmlrc2 0.1.7 [48] and the Illumina paired-end reads.

Purge_dups 1.2.6 [49] was used to remove haplotigs, and the quality of the final assembly was evaluated using yak 0.1 [50] and BUSCO 5.3.2 [51] with the eudicots_odb10 dataset.

#### 4.3.2. cT-DNA Annotation

*N. noctiflora* contigs containing cT-DNA sequences were identified through mapping the reference cT-DNA sequences to the *N. noctiflora* genome using minimap2 2.24 [52,53]. BLAST analysis on these contigs was then performed using blastx (NCBI, http://www.ncbi.nlm.nih.gov, accessed on 2 June 2023) using the ‘non-redundant protein sequences (nr)’ database, with the ‘Organism’ parameter limited for the *Rhizobium/Agrobacterium* group (taxid:227290).

### 4.4. PCR

#### 4.4.1. Reverse Transcription

Before cDNA synthesis, the amount of RNA for different tissues was normalized. For each sample, cDNA was synthesized with 330 ng of total RNA using an iScript cDNA Synthesis Kit (Bio-Rad, Hercules, CA, USA) according to the manual. The cDNA synthesis reaction (20 μL) was performed in 0.2 mL tubes on a Bio-Rad C1000 Thermal Cycler apparatus according to the following program: 5 min at 25 °C; 20 min at 46 °C; and 1 min at 95 °C.

#### 4.4.2. Real-Time qPCR

Real-time qPCR reactions (20 μL) were performed in optical 8-Tube strips with a Bio-Rad CFX96 apparatus using iTaq Universal SYBR Green Supermix 2× ((both from Bio-Rad), Hercules, CA, USA) and gene-specific primers. *N. noctiflora* glyceraldehyde 3-phosphate dehydrogenase gene (*gapdh*) was used as a reference for the equalization of RNA levels.

Primers used for RT-qPCR are listed in Table 2. All primers were synthesized by Evrogen (Moscow, Russia). The PCR was carried out according to the following program: 5 min at 94 °C; 30 cycles of 20 s at 94 °C, 20 s at 60 °C, and 30 s at 72 °C; followed by 10 min at 72 °C.

Quantification was performed in triplicate. Relative expression levels were calculated using the 2^−ΔΔCT^ method [34] for each sample, and then the average values for three plants were calculated.

#### 4.4.3. Molecular Cloning

The coding sequence of *Nnacs1* was PCR-amplified using the primers Nnacs1_topoF and Nnacs1_topoR (Table 2). The amplification reactions (40 μL) contained a DNA sample, 20 μL of DreamTaq PCR Master Mix (2×) (Thermo Fisher, Waltham, MA, USA), and 10 pmole of each primer. PCR was carried out using a “Tertsyk” DNA amplifier (“DNA technology”, Moscow, Russia), according to the following program: 5 min at 94 °C; 35 cycles of 15 s at 94 °C; 30 s at 60 °C; and 60 s at 72 °C; followed by 10 min at 72 °C. The PCR product was cloned using a pENTR™/D-TOPO™ Cloning Kit (Thermo Fisher, Waltham, MA, USA), according to its manual, and then subcloned in pDest527 using LR Clonase™ II Plus enzyme mix (Thermo Fisher, Waltham, MA, USA), according to the provided protocol. Then, 0.5 mL tubes with 5 μL of the cloning mix were incubated at 25 °C for 16 h in a “Tertsyk” DNA amplifier. Recombinant plasmids were transferred into NiCo21(DE3) chemically competent E. coli cells according to the protocol of Inoue and co-authors [54]. Colony PCR for insert confirmation was conducted with the same Nnacs1_topoF and Nnacs1_topoR primers. Colonies with the expected insert were used for further experiments.

### 4.5. Opine Extraction and Identification

Transgenic cells of *E. coli* were cultivated on LB medium, supplemented with ampicillin 100 mg/lL, to an optical density of 0.5–0.6. Then an equal volume of medium with 1 mM IPTG (isopropylthio-β-galactoside) (Thermo Fisher, Waltham, MA, USA), 0.02 M arabinose, 0.02 M glucose, and 0.02 M sucrose was added to induce the transgene expression and opine synthesis for 4 h. After that, the cells were pelleted via centrifugation for 10 min at 3700× *g* rpm. The original strain was used as a control. The medium without IPTG was used as a control for the cultivation conditions [55]. Analysis of the chemical structure was performed for the major compound, characteristic only for the transgenic strain, cultivated on a medium with IPTG.

Analytical TLC was performed on silica gel Kieselgel 60 F_254_ plates (Merck, Darmstadt, Germany). The plates were eluted with 6% boric acid, 60% acetic acid, ethanol, acetone, and ethyl acetate (10:15:20:60:60). The spots were visualized through spraying with the reagent anisaldehyde–H_2_SO_4_, followed by heating at 120 °C for 2 min. HPLC-MS was performed on a Shimadzu LCMS-IT-TOF instrument (Kyoto, Japan), according to the method described by Padilla et al. (2021) [46]. Acetonitrile and acidified water (0.4% formic acid) were used as the eluent at a flow rate of 0.3 mLl/min. The gradient started with 100% acidified water over 3 min and then ramped up to 100% acetonitrile over 4 min, maintained for 1 min before returning to initial conditions, and then held to equilibrate the column after collection (2 min). The injection volume of each sample was 5 μLl. Electrospray ionization was used in positive and negative ionization modes, with a drying gas (N_2_) flow of 1.5 L/min. Detection was carried out in the following voltage range: −2.5, −1.5, −1, +1.5, +2, and +3 kV.

MS/MS detection was performed using a UHR-TOF Mass Spectrometer MaXis from Bruker Daltonics (Billerica, MA, USA), all equipped with an electrospray ionization (ESI) source operating in positive- and negative-ion mode. MS data were collected over a range of m/z from 100 to 1600.

^31^P and ^1^H NMR spectra were recorded on an NMR spectrometer, the Bruker Avance III HD 400 III HD, at 161 MHz and 100.63 MHz, respectively, in D_2_O solutions. The Heteronuclear Multiple Bond Correlation (HMBC) method was used to interpret ^31^P and ^1^H NMR spectra. The full NMR spectra are given in the Appendix A.

## Figures and Tables

**Figure 1 plants-12-03787-f001:**
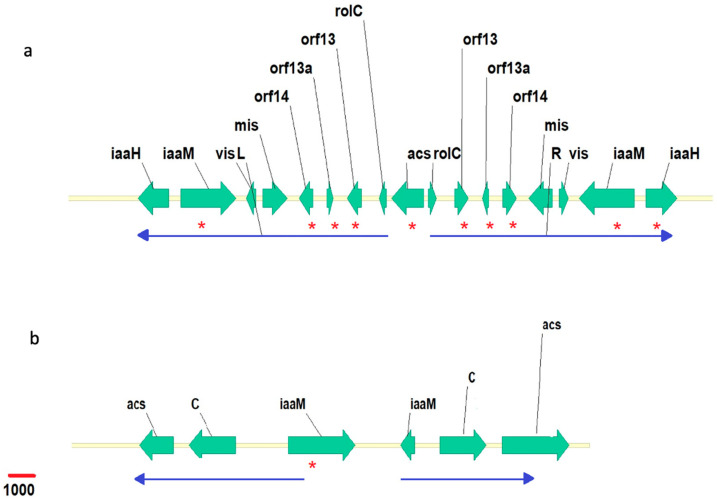
Maps of *Nn*T-DNA1 (**a**) and *Nn*T-DNA2 (**b**). Green arrows indicate sequences homologous to *Agrobacterium* T-DNA genes. Asterisks indicate intact ORFs. Thin blue arrows mark inverted repeats. The length of the red line corresponds to 1000 nucleotides.

**Figure 2 plants-12-03787-f002:**
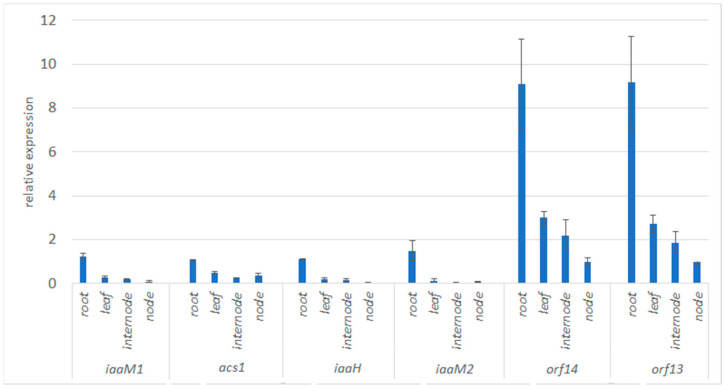
Relative expression of genes located on *Nn*T-DNA1 and *Nn*T-DNA2. This figure shows the relative presence of mRNA in different organs of *N. noctiflora*. The expression levels are shown relative to the expression level of the *gapdh* housekeeping gene.

**Table 1 plants-12-03787-t001:** cT-DNA genes detected in the *N. noctiflora* genome.

cT-DNA (Contig)	cT-DNA Gene (Homolog)	Copy Number	Intact	Positions	Identity Level to Proteins from NCBI		Similarity Level between Two Copies of the Gene, %
					% Identity	Organism and Protein ID	
NnT-DNA1 (contig_14451)	*iaaH*	2	−	144,696-145,941	89	*Agrobacterium rhizogenes*, WP_174054129.1	98
			+	165,221-166,483	89		
	*iaaM*	2	−	146,971-148,638	95	*Agrobacterium rhizogenes*, WP_174054254.1	99
			+	162,525-164,741	93		
	*vis*	2	−	149,083-149,457	94	*Agrobacterium rhizogenes*, WP_174054130.1	94
			−	161,720-162,080	91		
	*mis*	2	−	149,747-150,729	77	*Agrobacterium rhizogenes*, WP_010900210.1	95
			−	160,472-161,430	79		
	*orf14*	2	+	151,210-151,761	74	*Agrobacterium rhizogenes*, WP_174075805.1	99
			+	159,440-159,991	75		
	*orf13a*	2	+	152,340-152,576	50	*Agrobacterium rhizogenes*, ABS11827.1	100
			+	158,624-158,860	50		
	*orf13*	2	+	153,153-153,710	75	*Agrobacterium rhizogenes*, WP_174075804.1	99
			+	157,490-158,047	75		
	*rolC*	2	−	154,449-154,727	85	*Agrobacterium rhizogenes*, WP_174075803.1	99
			−	156,419-156,754	70		
	*acs*	1	+	154,937-156,241	82	*Agrobacterium rhizogenes*, GAJ95539.1	-
NnT-DNA2 (contig_30643)	*acs*	2	−	188,298-189,450	87	*Agrobacterium rhizogenes*, WP_174054263.1	96
			−	200,609-202,908	86		
	*rolC*	2	−	189,968-191,539	76	*Agrobacterium rhizogenes*, WP_174054195.1	99
			−	198,522-200,077	74		
	*iaaM*	2	+	193,348-195,616	75	*Agrobacterium rhizogenes*, WP_174054196.1	97
			−	197,150-197,650	75		

**Table 2 plants-12-03787-t002:** Sequences of primers.

Name	Sequence	Purpose of Use
Nglgapdh-F	5′-GGTGCCAAGAAGGTTGTGAT-3′	RT-qPCR
Nglgapdh-R	5′-CAAGGCAGTTGGTAGTGCAA-3′	
Nn1acs-F	5′-AAGAGGAGGGCATCACGTTG-3′	
Nn1acs-R	5′-TCGAGACCAATCCAAACGCA-3′	
Nn2acs-F	5′-TGATCTGCCATGAATGCCCT-3′	
Nn2acs-R	5′-GCACAGCCGATTAGAGTGGT-3′	
NnOrf13-F	5′-AGGCTCAGCTTATGAATGTGG-3′	
NnOrf13-R	5′-CAGCTCCATTTCCGTCTCAT-3′	
NnOrf13a-F	5′-GGACTTTGCCCGGAGATCGCT-3′	
NnOrf13a-R	5′-ACCGCTCCATTGGCTATGCTCA-3′	
NnOrf14-F	5′-GGACCTCGATCAGACTGTGAC-3′	
NnOrf14-R	5′-CGGCTGAATTGCTACTGTTG-3′	
Nn2iaaM-F	5′-AACGCAAATGTAGCCGAGGA-3	
Nn2iaaM-R	5′-GCTTGATCGTCGCCTGGATA-3′	
NniaaM-F	5′-CGGCTCTCGGACACTAAGTA-3′	
NniaaM-R	5′-CCATAAGCCACCATCTCAA-3′	
Nn2iaaH-F	5′-TGCATGACTGAGCAATCCTG-3′	
Nn2iaaH-R	5′-TGCTTTAAAGGGAATATTGC-3′	
Nnacs1_topoF	5′-CACCATGTACAATTGGGGTGAAGGAGTACA	molecular cloning
Nnacs1_topoR	5′-CTGCTATATGGCCATGTCATCTTCTAATC	

## Data Availability

Illumina and Oxford Nanopore reads are available from the National Center for Biotechnology Information Short Read Archive (SRA) under accession PRJNA982588. The genome assembly has been deposited at DDBJ/ENA/GenBank at https://www.ncbi.nlm.nih.gov/datasets/genome/GCA_032618575.1. accessed on 1 November 2023 (accession number JASVEL000000000). Cellular T-DNA sequences are available from the National Center for Biotechnology Information Nucleotide under accessions OR134233.1 (*Nn*T-DNA1) and OR134234.1 (*Nn*T-DNA2)

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
