# Peer review of "Nicotiana noctiflora Hook. Genome Contains Two Cellular T-DNAs with Functional Genes"

_plants, 2023, doi:10.3390/plants12223787_

Round 1

Reviewer 1 Report

Comments and Suggestions for Authors

Please address the following minor comments:

1. Figure 1 is not clear and readable for readers. Please redraw it in a more scientific and readable. 

2. Line 138: What does mean functional analyses for ORF? 

3. Line 140: Why gapdh has been used as a reference gene?

4. Line 141: Why? They should show it here.

5. Figure 2: This chart is unclear. The colors are not selected in proper. Moreover, statistical analysis should be carried out on these data to reveal significance.

6. Line 165: Naturally transgenic plants have become quite common, is nonsense. Please clarify.

7. Line 223: DNA isolation. Why RNase A has not been used to remove the extra RNAs from DNA extraction, while the extra RNAs cause to fault result.

Author Response

The authors express their deep gratitude to the reviewer for his careful reading of the manuscript and valuable recommendations.

Our responses to the reviewer's comments are presented below. Changes in the text of the article are marked in red.

  1. Figure 1 is not clear and readable for readers. Please redraw it in a more scientific and readable. 

Figure 1 is in the same style as previously published cellular T-DNA maps of natural GMOs. ( Matveeva, T.V., Otten, L. Widespread occurrence of natural genetic transformation of plants by Agrobacterium. Plant Mol Biol 101, 415–437 (2019). https://doi.org/10.1007/s11103-019-00913-y; Matveeva, T. (2021). New naturally transgenic plants: 2020 update. Biological Communications, 66(1), 36–46. https://doi.org/10.21638/spbu03.2021.105)

To make it more understandable we have added more detailed legend.

  1. Line 138: What does mean functional analyses for ORF? 

The incorrect phrase has been changed in the text

  1. Line 140: Why gapdh has been used as a reference gene?

It was used because its expression level was stable in our experimental conditions. We added this sentence to the text.

  1. Line 141: Why? They should show it here.

Orf13a is not shown on the diagram, since its expression was lower than reference gene in all studied tissues. This sentence is added to the text.

  1. Figure 2: This chart is unclear. The colors are not selected in proper. Moreover, statistical analysis should be carried out on these data to reveal significance.

This figure was corrected. Expression analysis was performed by 2-ΔΔCT method [43]. It is common method for gene expression analysis.

  1. Line 165: Naturally transgenic plants have become quite common, is nonsense. Please clarify.

Rephrased as follows:

In recent years, many natural GMOs have been described [15].

  1. Line 223: DNA isolation. Why RNase A has not been used to remove the extra RNAs from DNA extraction, while the extra RNAs cause to fault result.

We used the TruSeq Nano DNA LT kit for Illumina library preparation.  It is based on the process of fork-shaped dsDNA adaptor ligation to the adenylated dsDNA using T4 DNA ligase with subsequent PCR amplification. The ligation and amplification are efficient enough with RNA existing in the DNA sample. So, RNA contaminants, if any, are not converted into DNA libraries and cannot cause faulty results. To the best of our knowledge and experience and according to the manufacturer's protocol, the only problem with RNA contamination is the possibility of DNA concentration overestimation when using certain measuring methods. This may cause low DNA input and low library output, as a result. However, this was not the case.   

The high molecular weight genomic DNA for long-read sequencing was extracted using Promega Wizard HMW DNA Extraction Kit (It contains RNase A).

Reviewer 2 Report

Comments and Suggestions for Authors

Dear authors

Your manuscript describe interesting research data, which are not only important from the biological information point of view, but also to shade additional light on the genetic transformation controversy.

 However, the text, particularly the Introduction, needs to be completely rewritten

The English language needs to be revised and  the writing stylistically improved.

Main issues:

The genome assembly is still not publicly available and its address in the ncbi need to be included in the text. The same for the two ct-DNA loci (e.g. https://www.ncbi.nlm.nih.gov/nuccore/OR134233.1.)

A non-specialized reader needs to be informed that the T-DNA genes present in Agrobacterium are already ready for expression in plant (eukaryotic) cells,

Some very few examples of passages that need to be improved .

 L38-39. Opines can be metabolized as a source of  carbon and nitrogen for the nutrition of the same “Agrobacterium''. Please rephrase to make the writing clearer  to a  nonspecialized reader (e.g. why “same”? Eventually the same species/strain?). Please remove the quotation marks.

Line 41 “transferring them in a series of sexual generations” Is this series limited? Please rephrase (maybe using “progeny (ies) ?)  

 L42-44.  Most likely, they are descendants of plants regenerated from transgenic tissues, since it  has been experimentally demonstrated that hairy roots can regenerate into fertile plants  [10-12]. Rephrase to make the writing clearer and more exact (e.g. “demonstrated that fertile plants can be regenerated from hairy roots”). Actually not all hairy roots became a new plant.

 L52- 53section Tomentosae and occur  in different combinations among different species of the section Tomentosae.” Obs. This passage shows lack of attentive revision and correction of  the text.

 L- 54-55  etc.  “Nicotiana tomentosiformis (N. tomentosiformis), an ancestor of Nicotiana tabacum (N. tabacum) i.e., tobacco.” The first use of the full name followed by an abbreviation can make sense, but after this the repetition of “Nicotiana”  for the other species of the same genus do not make sense. Form here on the text is very badly organized and hard to be read. Please reformulate.  You are talking about two different sections. Please divide the text very clearly in two parts. Notice the  repeated “are in” substitute and say it in different ways.

 L 162-163 “Thus, the structure of the compound being studied is most similar to agrocinopine A.” Please rephrase, this sounds weird, as a final report from an external lab.   

 REFERENCES: Checkl very carefully. For example [11] has no publication year.

 Material and Methods

 Please remove the reference to the “clipped tips”. Is very common, and a question of good sense.

Best Regards

Comments on the Quality of English Language

The text, particularly the Introduction, needs to be completely rewritten

The English language needs to be revised and  the writing stylistically improved.

Author Response

The authors express their deep gratitude to the reviewer for his careful reading of the manuscript and valuable recommendations.

Our responses to the reviewer's comments are presented below. Changes in the text of the article are marked in green.

The genome assembly is still not publicly available and its address in the ncbi need to be included in the text. The same for the two ct-DNA loci (e.g. https://www.ncbi.nlm.nih.gov/nuccore/OR134233.1.)

  • NCBI Acc. Numbers for cT-DNAs are included in the text. We have requested the release of all the data and of the genome. It should be available at https://www.ncbi.nlm.nih.gov/bioproject/PRJNA982588soon

A non-specialized reader needs to be informed that the T-DNA genes present in Agrobacterium are already ready for expression in plant (eukaryotic) cells,

  • corrected

Some very few examples of passages that need to be improved .

 L38-39. Opines can be metabolized as a source of  carbon and nitrogen for the nutrition of the same “Agrobacterium''. Please rephrase to make the writing clearer  to a  nonspecialized reader (e.g. why “same”? Eventually the same species/strain?). Please remove the quotation marks.

·         Quotation marks were used to show, that term Agrobacterium is used in a broad sense. We have changed to Agrobacterium sensu lato

Line 41 “transferring them in a series of sexual generations” Is this series limited? Please rephrase (maybe using “progeny (ies) ?)  

  • rephrased

 L42-44.  Most likely, they are descendants of plants regenerated from transgenic tissues, since it  has been experimentally demonstrated that hairy roots can regenerate into fertile plants  [10-12]. Rephrase to make the writing clearer and more exact (e.g. “demonstrated that fertile plants can be regenerated from hairy roots”). Actually not all hairy roots became a new plant.

  • rephrased

 L52- 53 “section Tomentosae and occur  in different combinations among different species of the section Tomentosae.” Obs. This passage shows lack of attentive revision and correction of  the text.

  • rephrased

 L- 54-55  etc.  “Nicotiana tomentosiformis (N. tomentosiformis), an ancestor of Nicotiana tabacum (N. tabacum) i.e., tobacco.” The first use of the full name followed by an abbreviation can make sense, but after this the repetition of “Nicotiana”  for the other species of the same genus do not make sense. Form here on the text is very badly organized and hard to be read. Please reformulate.  You are talking about two different sections. Please divide the text very clearly in two parts. Notice the  repeated “are in” substitute and say it in different ways.

  • We divided the text in two parts.

 L 162-163 “Thus, the structure of the compound being studied is most similar to agrocinopine A.” Please rephrase, this sounds weird, as a final report from an external lab.   

  • rephrased

 REFERENCES: Check very carefully. For example [11] has no publication year.

  • It has publication year. One reference added to the list.

 Material and Methods

 Please remove the reference to the “clipped tips”. Is very common, and a question of good sense.

  • removed

Round 2

Reviewer 2 Report

Comments and Suggestions for Authors

Dear Authors

This an interesting manuscript, from the scientific point of view and social perception of the GMOs.

However, in my opinion, some issues remain to be solved.

Major issue

The genome assembly still is not available (e.g. NCBI) for consultation.

How can a reader confirm your statements?

Minor issues

The evidence that authors have not made a real effort for the correction of the manuscript but only focused on the examples (not all cases) of needed corrections is, for example, the maintenance of the concept (???) “Agrobacterium”.

Notice that the discrepancy in style between the “Introduction” and “Discussion” is very clear. Please  uniformize.

Examples (not all) of the needed additional corrections.

1) Please revise again the text carefully. Particularly the sequence of “are” in lines 54-57.

2) Rephrase: “The function of some genes has been demonstrated at the RNA level, but also the level of the final metabolite”

3) Based on genomic sequencing data, the presented work reconstructed the structure of N. noctiflora cT-DNA, demonstrated the tissue-specific expression of its intact genes, and described the structure of the agrocinopine encoded by the Nn 79 acs1 gene.

Example of  other minor corrections (or clarifications).

1) “aseptic N. noctiflora plants” – Are these plants growing in vitro? Please, clarify.

Comments on the Quality of English Language

The English Language needs to be improved.

Author Response

The authors express their deep gratitude to the reviewer for his careful reading of the manuscript and valuable recommendations.

Our responses to the reviewer's comments are presented below. The new changes in the text of the article are marked in blue.

  1. The genome assembly is available under accession JASVEL000000000
  2. Corrections in the text:

The evidence that authors have not made a real effort for the correction of the manuscript but only focused on the examples (not all cases) of needed corrections is, for example, the maintenance of the concept (???) “Agrobacterium”.

  - The taxonomy of the genus Agrobacterium has been revised in recent years. For this reason, the term has been used in a broad sense with appropriate punctuation. In the current version, we have removed «» from everywhere, replacing them with synonymous expressions.

Notice that the discrepancy in style between the “Introduction” and “Discussion” is very clear. Please  uniformize.

- we have corrected the text in the discussion

Examples (not all) of the needed additional corrections.

1) Please revise again the text carefully. Particularly the sequence of “are” in lines 54-57.

2) Rephrase: “The function of some genes has been demonstrated at the RNA level, but also the level of the final metabolite”

- Rephrased

3) Based on genomic sequencing data, the presented work reconstructed the structure of N. noctiflora cT-DNA, demonstrated the tissue-specific expression of its intact genes, and described the structure of the agrocinopine encoded by the Nn 79 acs1 gene.

- corrected

Example of  other minor corrections (or clarifications).

  • “aseptic N. noctiflora plants” – Are these plants growing in vitro? Please, clarify.
  • In our case is more important, that these plants are aseptic (without any bacteria). In vitro could be understood in different ways (in Petri dish or in green house) In green house bacterial contamination is possible. So it will be in vitro, but not aseptic.
  • We changed the text to «aseptic in vitro noctiflora plants»

Round 3

Reviewer 2 Report

Comments and Suggestions for Authors

Dear authors

Please refer to the access to the genome assembly as 

https://www.ncbi.nlm.nih.gov/datasets/genome/GCA_032618575.1/

Best Regards

Author Response

Thank you very much for the careful reading of the manuscript and your comment.

We corrected the text according to the recommendation and marked it in purple.